# Crop and Segment: Efficient Whole body pan-cancer segmentation

Yinyin Luo[†], Yue Liu[†], Wenbin Liu, Jingheng Dai, Xunliang Xiao, and
Gang Fang[*]

Institute of Computing Science and Technology, Guangzhou University, Guangzhou,
510006, China
[†]Co-first authors
[*]Corresponding author
gangf@gzhu.edu.cn

**Abstract.** Despite significant advancements in deep learning models for medical segmentation, the detection and segmentation of tumors, particularly through whole-body scans, remain challenging. To address this issue, we explored the application of nnUNet for whole-body tumor segmentation in CT scans, proposing a more precise cropping strategy and introducing an organ-interference segmentation approach to effectively enhance segmentation efficiency. Experiments on the MICCAI FLARE 2024 dataset demonstrated significant improvements in both segmentation accuracy and efficiency. Our method achieved an average organ Dice Similarity Coefficient (DSC) of 10.47% and a Normalized Surface Dice (NSD) of 7.98% on the public validation set. In the FLARE 2024 Task 1 online validation, the method achieved an average organ Dice Similarity Coefficient (DSC) of 17.08%, a Normalized Surface Dice (NSD) of 7.42% and the average running time and area under GPU memory-time curve were 19.89s and 45688 MB, respectively. The code is available at https://github.com/lay-john/FLARE24-Task1.

**Keywords:** Semi-supervised · Deep learning · Tumour segmentation .

## 1 Introduction

In recent years, deep learning models have significantly advanced the field of medical image analysis, particularly in medical segmentation. Notable models such as U-Net [20], which introduced a novel architecture for biomedical image segmentation, and its extended versions like nnU-Net [10], have demonstrated substantial improvements in various segmentation tasks. These advancements have facilitated enhanced performance in segmenting organs and pathological regions across numerous medical imaging modalities. However, despite these breakthroughs, the accurate detection and segmentation of tumors in whole-body scans continue to pose significant challenges. Tumors can vary greatly in shape, size, and appearance across different anatomical regions, making their segmentation inherently complex. Additionally, whole-body scans involve high-resolution

3D CT images that are both computationally intensive and resource-demanding. This complexity is exacerbated by the need to process large volumes of data while maintaining high accuracy and speed. Recent studies have highlighted the difficulties in extending segmentation models to whole-body scans. For instance, the work by Liu et al. (2018) on the DeepMedic model [12] demonstrated improved performance in brain tumor segmentation, but challenges remain in generalizing these methods to whole-body tumor detection. Similarly, the work by Zhou et al. (2019) on 3D U-Net [28] showed promising results for organ segmentation but highlighted the need for further improvements in handling the variability of tumors across the body. To address these challenges, various approaches have been explored. For example, the study by Yang et al. (2021) proposed a multi-scale approach to handle the variability in tumor sizes and shapes [26]. Despite these advancements, achieving accurate and efficient whole-body tumor segmentation remains a critical and ongoing challenge in the field of medical imaging.Among these methods, it is highly noted that nnUNet has consistently demonstrated outstanding performance in different medical image segmentation tasks.

Our study builds on these advancements by leveraging the nnUNet framework, known for its robust and flexible design in medical image segmentation. We propose novel strategies to enhance the efficiency and accuracy of whole-body tumor segmentation, including more precise cropping methods and refined post-processing techniques. This approach aims to address the inherent challenges associated with high-resolution 3D CT scans, ultimately improving the performance and applicability of segmentation models in clinical practice.To tackle these challenges, we focused on leveraging the nnUNet framework, a state-of-the-art model known for its robustness and flexibility in medical image segmentation. nnUNet has shown promising results in various medical imaging tasks.

Specifically, we propose a cropping strategy based on the largest connected component, which retains the largest connected region in CT images by evaluating voxel intensity relationships. The boundaries for cropping are then determined based on the content of the retained largest connected component, thereby achieving a more precise delineation of the Region of Interest (ROI). Furthermore, During our experiments, we observed that most tumor labels in the dataset were only partially annotated. Due to the complexity of the whole-body regions, the trained model frequently misclassified most organ areas as tumors, even though the actual tumor regions in each CT scan occupy only a small portion. To address this, we introduced an organ-interference segmentation approach. By jointly training the model on both tumors and organs, we enforced the model to focus more on the tumor regions during training, thus helping it to eliminate most of the misclassified organ areas without significantly impacting the accuracy of tumor region predictions. The detailed methodology is described in Section 4.1.

To summarize, this paper presents a precise cropping strategy based on the largest connected component and introduced an organ-interference segmentation approach to address MICCAI FLARE 2024 Challenge Task 1. The main contributions are as follows: (1) To achieve more accurate cropping regions and

improve inference speed, we propose a precise cropping strategy based on the largest connected component, which enhances both inference speed and segmentation accuracy. (2) To reduce false positives in tumor segmentation, we further introduce an organ-interference segmentation approach. Experimental results demonstrate the effectiveness of this approach in improving model performance.

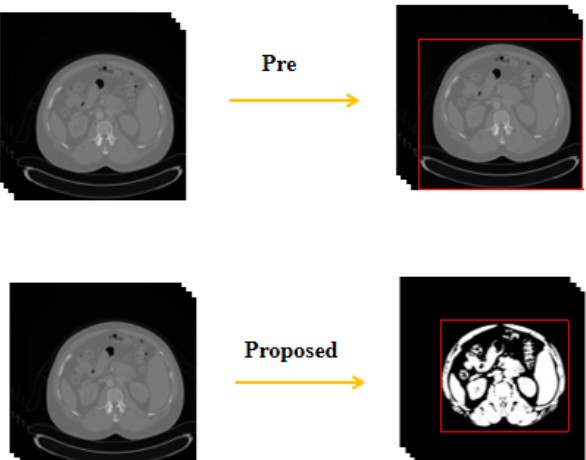

**Fig. 1.** "Pre" refers to the previous default cropping method, while "Proposed" denotes the cropping strategy based on the largest connected component that we have introduced in this study.

## 2   Method

### 2.1   Preprocessing

– **Resample and normalization:**  We resample the pixel spacing to (1.6, 1.2859, 1.2859) for all cases, and clip the pixel value based on the Hounsfield units to [-160, 240], and normalize all the cases in [0, 1] to ensure data stability and consistency.
– **Data augmentation:** In order to prevent the model from over-fitting, data augmentation is used in this study. The augmentation approaches of nnU-Net methodology have been utilized.

## 2.2   Proposed Method

Specifically, the primary focus of this work is on preprocessing and the organ-interference. Therefore, we utilize the default nnUNet architecture as our model for training. We use the provided FLARE 2024 Task 1 dataset, which comprises over 10,000 CT scans covering various whole-body cancer types. The dataset includes 5,000 partially annotated scans and 5,490 unannotated scans. In the partially annotated dataset, only the primary lesions are labeled, while other lesions, such as metastatic ones, may remain unlabeled. For our model training, we use only the 5,000 partially annotated scans.The first contribution of this paper is the proposal of a cropping strategy based on the largest connected component. This strategy retains the largest connected region in CT images by evaluating voxel intensity relationships. The cropping boundaries are then determined based on the content of the retained largest connected component, resulting in more precise and smaller cropping regions. Another contribution is the introduction of an organ-interference segmentation approach. By jointly training the model on both tumors and organs, we enforced the model to focus more on the tumor regions during training, thus helping it to eliminate most of the misclassified organ areas without significantly impacting the accuracy of tumor region predictions.

**A cropping strategy based on the largest connected component:** First, we obtain a mask by thresholding the original CT voxel values greater than 50. Then, the largest connected component is preserved within this mask, resulting in a refined mask containing only the largest connected region. Based on this refined mask, we determine the bounding box for cropping and subsequently perform the cropping operation.As shown in Fig. 1., it can be seen that compared with previous cropping methods, our cropping strategy can significantly reduce the cropped area without affecting our subsequent segmentation.

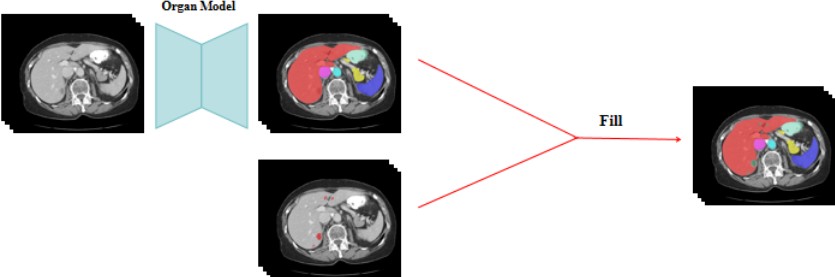

**Fig. 2.** Generate final organ and tumor hybrid labels using organ models and original labels.

**An organ-interference segmentation approach:** We utilized the abdominal CT data from FLARE 2022 [16] to train an organ model. This model was then applied to generate labels for abdominal organs on 5,000 annotated CT scans, resulting in pseudo-labels for the abdominal organs. The tumor labels from the provided annotations were then integrated into the generated pseudo-labels, forming the final training labels, as illustrated in Fig. 2. During training, these final labels were sampled with a certain probability during data loading. If a label was selected, we extracted a patch centered on the selected label. We increased the probability of selecting tumor labels by approximately five times compared to other labels, while the probabilities for the remaining organ labels were kept equal. The training process employed a weighted DiceLoss and CELoss. For the weight w, we set the tumor regions assigned a weight five times higher than the organ regions. This weighting scheme was designed to direct the model's focus toward more accurate tumor segmentation. Our loss function formula is as follows:

$$L_{Dice}(y, \hat{y}, w) = \sum_i^{c_0} w_i \left( 1 - \frac{2 \sum_{j=1}^{N} y_j^i \hat{y}_j^i}{\sum_{j=1}^{N} y_j^i + \hat{y}_j^i} \right) \tag{1}$$

$$L_{CE}(y, \hat{y}, w) = \sum_i^{c_0} w_i \left( -\frac{1}{N} \sum_{j=1}^{N} \left( y_j^i \log(\hat{y_j^i}) + (1 - y_j^i) \log(1 - \hat{y_j^i}) \right) \right) \tag{2}$$

$$L_{Seg} = \alpha_{dc} \cdot L_{Dice}(y, \hat{y}, w) + \alpha_{ce} \cdot L_{CE}(y, \hat{y}, w) \tag{3}$$

Where the $y$ and $\hat{y}$ mean the ground truth and the predicted probability, respectively, and $N$ is the number of pixels. $\alpha_{dc}$ and $\alpha_{ce}$ are the hyperparameters to balance the contribution of DiceLoss and CELoss. $\alpha_{dc}$ and $\alpha_{ce}$ are set to 0.5 in this study.

### 2.3 Post-processing

We remove the organ labels from the inferred results and only retain the tumor labels, as shown in Fig. 3. In addition, no post-processing operations are performed.

## 3 Experiments

### 3.1 Dataset and evaluation measures

The segmentation targets cover various lesions. The training dataset is curated from more than 50 medical centers under the license permission, including TCIA [3], LiTS [2], MSD [22], KiTS [7,9,8], autoPET [6,5], TotalSegmentator [23], and AbdomenCT-1K [18], FLARE 2023 [17], DeepLesion [25], COVID-19-CT-Seg-Benchmark [15], COVID-19-20 [21], CHOS [11], LNDB [19], and

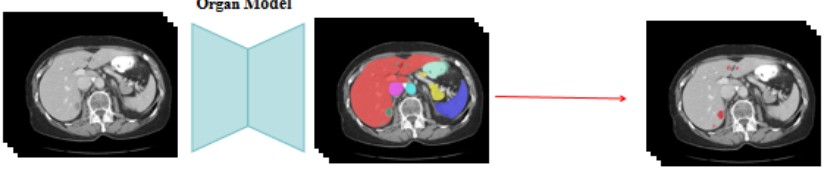

**Fig. 3.** Processing of inferred labels.

LIDC [1]. The training set includes 4000 abdomen CT scans where 2200 CT scans with partial labels and 1800 CT scans without labels. The validation and testing sets include 100 and 400 CT scans, respectively, which cover various abdominal cancer types, such as liver cancer, kidney cancer, pancreas cancer, colon cancer, gastric cancer, and so on. The lesion annotation process used ITK-SNAP [27], nnU-Net [10], MedSAM [13], and Slicer Plugins [4,14].

The evaluation metrics encompass two accuracy measures—Dice Similarity Coefficient (DSC) and Normalized Surface Dice (NSD)—alongside two efficiency measures—running time and area under the GPU memory-time curve. These metrics collectively contribute to the ranking computation. Furthermore, the running time and GPU memory consumption are considered within tolerances of 45 seconds and 4 GB, respectively.

### 3.2   Implementation details

**Environment settings** The development environments and requirements are presented in Table 1.

**Table 1.** Development environments and requirements.

| System | Ubuntu 22.04 LTS or Windows 10 |
|---|---|
| CPU | Intel(R) Core(TM) i9-10900X CPU@3.70GHz |
| RAM | 4×32GB; 2933MT/s |
| GPU (number and type) | NVIDIA GeForce RTX™3090 24G |
| CUDA version | 12.1 |
| Programming language | Python 3.9.16 |
| Deep learning framework | torch 2.1.0, torchvision 0.16.0 |
| Specific dependencies | nnU-Net 1.7.0 |
| Code | https://github.com/lay-john/FLARE24-Task1 |

**Training protocols** During the training phase, we set the batch size to 2 and randomly select all samples within each epoch. For each sample, we perform random patch cropping with patch sizes of (96, 128, 160). As for the optimizer, we utilize AdamW with a learning rate of 1e-2 and a weight decay of 1e-5. The learning rate updating follows the default mechanism of AdamW. Additional details are presented in Table 2.

**Table 2.** Training protocols.

| | |
|---|---|
| Network initialization | |
| Batch size | 2 |
| Patch size | 96×128×160 |
| Total epochs | 500 |
| Optimizer | AdamW with weight decay($\mu$ = 1e -5) |
| Initial learning rate (lr) | 0.01 |
| Lr decay schedule | halved by 200 epochs |
| Training time | 42.5 hours |
| Loss function | DiceLoss and CELoss |
| Number of model parameters | 30.8M[1] |
| Number of flops | 838.6116 KG[2] |
| $CO_2$eq | 3.91908 Kg[3] |

## 4  Results and discussion

### 4.1  Quantitative results on validation set

To conduct a more comprehensive ablation study of our proposed method, we performed quantitative experiments, as shown in Table 3.

**Table 3.** Result in Public Validation, Online Validation and Final Testing.

| Methods | Public Validation | | Online Validation | | Testing | |
|---|---|---|---|---|---|---|
| | DSC(%) | NSD(%) | DSC(%) | NSD(%) | DSC(%) | NSD (%) |
| Algorithm | 10.47 | 7.98 | 17.08 | 7.42 | 32.73 | 20.58 |

To visually demonstrate the impact of our method on inference speed, we conducted quantitative experiments on inference speed, as shown in Table 4. The length of the step is [7/8, 7/8, 7/8] times the window width for each axis.

**Table 4.** Overview of Ablation Experiment Results.

| Target | Base | | Proposed | |
|---|---|---|---|---|
| | DSC(%) | NSD(%) | DSC(%) | NSD(%) |
| Tumour | 3.97 | 1.98 | 10.47 | 7.98 |

To visually demonstrate the impact of our method on inference speed, we conducted quantitative experiments on inference speed, as shown in Table 5. The length of the step is [7/8, 7/8, 7/8] times the window width for each axis.

**Table 5.** Overview of Ablation Experiment Results on inference speed. Time is measured in seconds.

| Target | Average time(s) |
|---|---|
| Base | 9.89 |
| Proposed | 8.09 |

### 4.2   Qualitative results on validation set

In this section, we show the two good segmentation cases and two bad segmentation cases.

Good segmentation cases: Fig. 4 presents examples of good segmentation results. It can be observed that the segmentation performance of our method is nearly comparable to that of manual annotations. In comparison to the baseline, while the baseline can also segment the tumor regions, it exhibits inferior boundary delineation for tumors. This improved boundary accuracy is one of the key reasons why our method outperforms the baseline.

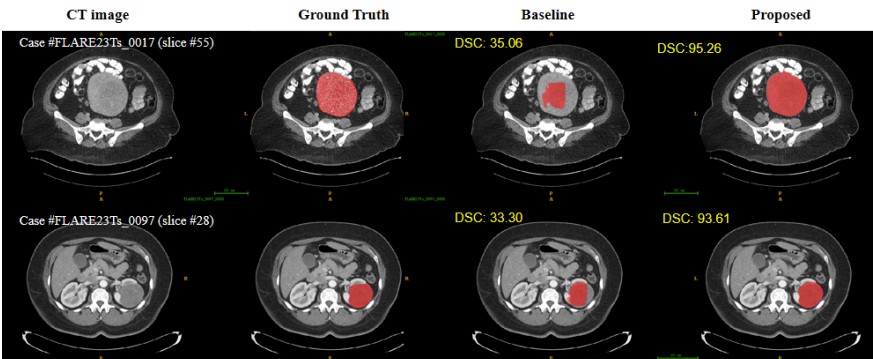

**Fig. 4.** Good segmentation cases from public validation set.

Bad segmentation cases: Fig. 5 presents examples of bad segmentation results. In the case of LNDb-0303, we observe that there is a small tumor region in the Ground Truth that our method failed to segment. Although the baseline was able to segment this region, it also incorrectly classified large additional areas as tumors, resulting in suboptimal segmentation performance. For LNDb-0312, where there is no tumor region in the Ground Truth, both our method and the baseline misclassified other regions as tumors. These two cases highlight that our method struggles with CT scans where there is either a very small tumor region or no tumor at all. This could be due to the model's inherent insensitivity to such regions or potentially incomplete training. Due to the summer break and other factors, the final model was trained on a single 3060 GPU for only 250 epochs. Another possible explanation is the absence of organs in these regions, leading to no organ interference, which might have contributed to the poor, or even highly inaccurate, segmentation performance in these areas.

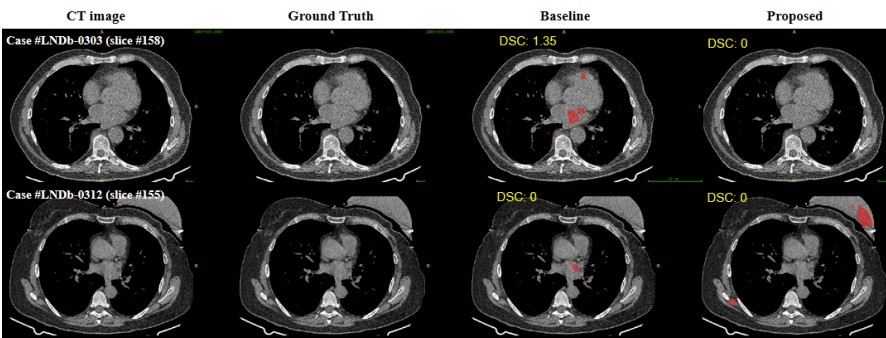

**Fig. 5.** Bad segmentation cases from public validation set.

### 4.3  Segmentation efficiency results on validation set

We quantitatively evaluate the segmentation efficiency of our model, as shown in Table 6. We also tested our model the false positive rate on the healthy CT scans and concluded that our method has the false positive rate of 0.875 on provided healthy CT scans.

### 4.4  Results on final testing set

The final testing results for our proposed method in the FLARE 2024 challenge are summarized in Table 7. The table presents the performance metrics of our method, including the Dice Similarity Coefficient (DSC), Normalized Surface Distance (NSD), inference time, and GPU memory usage. Each metric is reported with both the mean and standard deviation (Mean ± Std), as well as the median along with the first and third quartiles (Median (Q1, Q3)).

**Table 6.** Quantitative evaluation of segmentation efficiency in terms of the running them and GPU memory consumption. Total GPU denotes the area under GPU Memory-Time curve. Evaluation GPU platform: NVIDIA QUADRO RTX5000 (16G).

| Case ID | Image Size | Running Time (s) | Max GPU (MB) | Total GPU (MB) |
|---------|------------|------------------|--------------|----------------|
| 0001 | (512, 512, 55) | 6 | 4008 | 14146 |
| 0051 | (512, 512, 100) | 12 | 4660 | 43015 |
| 0017 | (512, 512, 150) | 12 | 4738 | 43835 |
| 0019 | (512, 512, 215) | 9 | 4194 | 26961 |
| 0099 | (512, 512, 334) | 7.5 | 4386 | 24930 |
| 0063 | (512, 512, 448) | 10.5 | 4589 | 30123 |
| 0048 | (512, 512, 499) | 9 | 4566 | 30352 |
| 0029 | (512, 512, 554) | 16.5 | 5142 | 60602 |

**Table 7.** Final testing results of the proposed method on the FLARE 2024 challenge.

| Metric | Mean ± Std | Median (Q1, Q3) |
|--------|------------|-----------------|
| DSC (%) | 32.73 ± 31.82 | 22.78 (0.00, 60.25) |
| NSD (%) | 20.58 ± 21.40 | 16.78 (0.00, 34.94) |
| Inference Time (s) | 19.89 ± 7.20 | 18.07 (14.10, 22.78) |
| GPU Memory (MB) | 45688.3 ± 14149.0 | 43234.0 (34802.5, 51978.5) |

### 4.5   Limitation and future work

In this study, we incorporated only abdominal organs to interfere with tumor segmentation. As a result, our method demonstrated good overall performance in tumor segmentation within the abdominal region. However, the performance in other regions was less satisfactory. In future work, we plan to explore the inclusion of additional organ labels to enhance tumor segmentation in non-abdominal areas. Furthermore, the accuracy of our model in delineating tumor boundaries requires further improvement, and we will also focus on refining boundary segmentation in future studies.

## 5   Conclusion

To tackle the challenging task of whole-body pan-cancer segmentation and improve the overall performance, this paper proposes a tumor segmentation method based on organ interference. To enhance inference speed, we also introduce a cropping strategy based on the largest connected component. Both quantitative and qualitative results demonstrate that the proposed method effectively and flexibly learns tumor information from the dataset. We validated our approach on the MICCAI FLARE 2024 challenge dataset, proving its efficacy in whole-body pan-cancer segmentation.

**Acknowledgements** The authors of this paper declare that the segmentation method they implemented for participation in the FLARE 2024 challenge has

not used any pre-trained models nor additional datasets other than those provided by the organizers. The proposed solution is fully automatic without any manual intervention. We thank all data owners for making the CT scans publicly available and CodaLab [24] for hosting the challenge platform.

## Disclosure of Interests

The authors declare no competing interests.

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

**Table 8.** Checklist Table. Please fill out this checklist table in the answer column.

| Requirements | Answer |
|---|---|
| A meaningful title | Yes |
| The number of authors ($\leq 6$) | 6 |
| Author affiliations and ORCID | Yes |
| Corresponding author email is presented | Yes |
| Validation scores are presented in the abstract | Yes |
| Introduction includes at least three parts: background, related work, and motivation | Yes |
| A pipeline/network figure is provided | 1 2 3 |
| Pre-processing | 3 |
| Strategies to improve model inference | 4 5 |
| Post-processing | 6 |
| The dataset and evaluation metric section are presented | 5 6 |
| Environment setting table is provided | 1 |
| Training protocol table is provided | 2 |
| Ablation study | 7 8 |
| Efficiency evaluation results are provided | 3 |
| Visualized segmentation example is provided | 4 5 |
| Limitation and future work are presented | Yes |
| Reference format is consistent. | Yes |