# OpenReview forum: "Crop and Segment: Efficient Whole body pan-cancer segmentation"
_MICCAI.org/2024/Challenge/FLARE — Submitted to FLARE 2024_

### Official Review · Reviewer_rpBm · 2025-03-02
**good method but writing is poor**

**Rating:** 5
**Confidence:** 5

**Review:**

There should be a space before the text e.g., U-Net [19] rather than U-Net[19]. No need to add (Ronneberger et al., 2015) after a reference.

Fig. 1 Please adjust all CT images to the proper window width and level (e.g., 400/40 for abdomen CT)
Fig. 3 Add explanations in the title

Table 2. Please fill out the correct numbers for the Number of flops and CO2eq
Table 3-5. Tables are too simple and formats are not consistent. If one table only has one row or one column, please directly describe the results with text.

Case-wise runtime analysis on the validation set is missed.

Typos and style: There are many typos or formatting issues, such as
“…mentation in CT scans, Proposing a more precise cropping strategy and
introducing an… “ P should be in lowercase
“Our loss function formula is as follows:
 Where”

The writing style in several sections is overly informal and does not align with the conventions of academic writing. Such as
“Pre is the previous default cropping method, Proposed is what we proposed…”

Please carefully read the whole paper and improve the writing quality to avod rejection in the last round review.

---

> ### Author Response · Authors · 2025-03-31
> **Response to Reviewer rpBm**
>
> Thank you for your valuable advice We have adjusted our paper in response to your comments.

---

### Official Review · Reviewer_fdaN · 2025-03-02
**Review of "Crop and Segment: Efficient Whole body pan-cancer segmentation"**

**Rating:** 6
**Confidence:** 5

**Review:**

This papaer proposes a cropping strategy and introduces an organ-interference segmentation approach to effectively enhance segmentation efficiency. The whole is not bad, but the writing details should be paid attention to.
1. Add a space before a reference, e.g., U-Net [19] rather than U-Net[19]. No need to add (Ronneberger et al., 2015) after a reference.
2. Please adjust all CT images to the proper window width and level (e.g., 400/40 for abdomen CT).
3. Add a space after punctuation marks, e.g., "as shown in Table 5. the length" rather than "as shown in Table 5.the length".
4. Note the use of the third person singular. For example, "remain" should be used instead of "remains" in Abstract.

---

> ### Author Response · Authors · 2025-03-31
> **Response to Reviewer fdaN**
>
> We sincerely thank Reviewer fdaN for the valuable comments and constructive feedback. Below are our point-by-point responses addressing the concerns raised.
>
> **Comment 1: Add a space before a reference, e.g., U-Net [19] rather than U-Net[19]. No need to add (Ronneberger et al., 2015) after a reference.**
>
> Thank you for your valuable comments, which have been revised accordingly in the paper.
>
> **Comment 2: Please adjust all CT images to the proper window width and level (e.g., 400/40 for abdomen CT).**
>
> Because task 1 is a whole-body segmentation task, all CT images are not uniformly adjusted to the same window width and window position.
>
> **Comment 3: Add a space after punctuation marks, e.g., "as shown in Table 5. the length" rather than "as shown in Table 5.the length".**
>
> Adjustments have been made for this issue.
>
> **Comment 4: Note the use of the third person singular. For example, "remain" should be used instead of "remains" in Abstract.**
>
> Modifications have been made in the paper.

---

### Official Review · Reviewer_8JAo · 2025-03-02
**Review of "Crop and Segment: Efficient Whole body pan-cancer segmentation"**

**Rating:** 7
**Confidence:** 5

**Review:**

The paper makes two key contributions. First, the cropping strategy based on the largest connected component refines the region of interest, enhancing both inference speed and segmentation accuracy. Second, the organ - interference segmentation approach reduces false positives in tumor segmentation by jointly training on tumors and organs. The comments are listed below:
(1) Please report the GPU consumption in the Abstract.
(2) What is the w in Eq. (1) and (2)?
(3) The authors said "alpha_KD is the hyperparameter of knowledge distillation." but the alpha_KD does not appear in any equation.
(2) The image quality could be improved. Vector graphics is highly recommended.
(3) Please rephrase the caption of Fig.1, Fig. 2, and Fig. 3 to describe the corressponding figures in detail.

---

> ### Author Response · Authors · 2025-03-31
> **Response to Reviewer 8JAo**
>
> Thank you for your valuable review comments. Below are our responses to the issues you raised:
>
> **Comment 1: Please report the GPU consumption in the Abstract.**
>
> GPU consumption is already reported in the summary.
>
> **Comment 2: What is the w in Eq. (1) and (2)?**
>
> The w has already been added in the paper.
>
> **Comment 3: The authors said "alpha_KD is the hyperparameter of knowledge distillation." but the alpha_KD does not appear in any equation.**
>
> The alpha_KD has been removed from the paper.
>
> **Comment 4: The image quality could be improved. Vector graphics is highly recommended.**
>
> Thank you for your suggestion, and we will strengthen this capacity in the future.
>
> **Comment 5: Please rephrase the caption of Fig.1, Fig. 2, and Fig. 3 to describe the corressponding figures in detail.**
>
> Thank you for your valuable suggestions for the caption of Figures 1, 2, and 3. We've revised the caption to provide a more detailed description of the corresponding chart.

---

### Decision · Program_Chairs · 2025-03-20

**Decision:**

Accept

**Comment:**

Please carefully address the reviewers' comments in the revision.

---

> ### Comment · Program_Chairs · 2025-03-31
>
> Sec 4.4 is not completed.
>
> Point-to-point response is not available.

---

> > ### Author Response · Authors · 2025-04-01
> > **Response to Program Chairs**
> >
> > Thank you for your feedback. We have carefully addressed all the reviewers' comments and revised the manuscript accordingly. A detailed response to each point is included in the revised submission, and the requested test results have been incorporated into the manuscript.

---

> ### Author Response · Authors · 2025-04-01
> **Response to Program Chairs**
>
> Thank you for your feedback. We have carefully addressed all the reviewers' comments and revised the manuscript accordingly. A detailed response to each point is included in the revised submission, and the requested test results have been incorporated into the manuscript.